# The Key Role of Empathy in the Relationship between Age and Social Support

**DOI:** 10.3390/healthcare11172464

**Published:** 2023-09-04

**Authors:** Paola Guariglia, Massimiliano Palmiero, Anna Maria Giannini, Laura Piccardi

**Affiliations:** 1Faculty of Human and Social Sciences, UKE—Kore University of Enna, 94100 Enna, Italy; 2Department of Communication Sciences, University of Teramo, 64100 Teramo, Italy; mpalmiero@unite.it; 3Department of Psychology, Sapienza University of Rome, 00158 Rome, Italy; annamaria.giannini@uniroma1.it (A.M.G.); laura.piccardi@uniroma1.it (L.P.); 4San Raffaele Cassino Hospital, 03043 Cassino, Italy

**Keywords:** well-being, affective support, community engagement, ageing, mental health, socio-emotional skills, empathy, socio-cognitive rehabilitation

## Abstract

Aging involves several changes depending on genetic and behavioral factors, such as lifestyle and the number and quality of social relationships, which in turn can be influenced by empathy. Here, the change in the perceived social support across the lifespan as a function of empathy was investigated, considering the mediating role of empathy after controlling for gender and education. In total, 441 people (18–91 years old) filled in the Italian short version of the Interpersonal Support Evaluation List (ISEL-12), the Social Support Questionnaire (SSQ6), as well as the Empathy Questionnaire (EQ), and the Reading the Mind in the Eyes test (RMET). The mediation analyses with ISEL-12 showed that age and the EQ fully mediated the relationship between age and appraisal, belonging, and tangible scores. Further, the EQ fully mediated only the relationship between age and SSQ6-People. These results showed that empathic skills are key in the relationships between age and social support. This suggests that empathy can trigger social support and, ultimately, well-being if stimulated across the lifespan, especially from a young age; this would help to form the socio-emotional competence across the years as a sort of cushion that can be useful in the older to fulfill active aging.

## 1. Introduction

Social support is defined as the provision of emotional, instrumental, or informational assistance or guidance [1]. Above all, it involves the perception that one is cared for, esteemed, and part of a mutually supportive social network, and consequently produces several positive effects on mental and physical health [2]. For example, perceived social support moderates the appraisal of threatening situations and enhances self-confidence to cope with new challenges [3,4,5]. Furthermore, the perception of support is a better predictor of health outcomes than the actual receipt of support [6,7,8]. When an individual is able to perceive the existence of social support, he/she feels a sense of belonging, increasing the capability to recognize self-worth [9,10,11,12].

Social support changes as age increases. Within the theoretical frame of the socioemotional selectivity theory (SST) [13], age differences in goals and time horizons influence social preferences and the composition of social networks, as well as cognitive processing. When social interactions, as well as the composition of social networks, change, especially older adults choose to spend more of their time with close others. Decreased relationships with age led to wonder about older adults being lonely and depressed [14]. However, other studies reported that this change in social networks reflects a proactive pruning process whereby older adults increasingly invest their limited time in relationships with close others [15]. Not only does the number of relationships change, but also the quality of social contacts. Relationships improve over time with experience and motivation, especially with children, spouses, and significant others [13]. 

Undoubtedly, supportive social relationships have positive effects on physical and mental health [15]. It is noteworthy that social networks are not unchanging throughout one’s life. Wrzus and colleagues [16] performed a meta-analysis of 277 studies and discovered some interesting trends. Firstly, social networks tend to increase during adolescence and early adulthood, followed by a decrease later on. Secondly, both personal and friendship networks tend to decrease throughout adulthood. However, the family network tends to remain stable in size. Lastly, certain networks such as colleagues and neighbors are only significant during specific age ranges. Summing up, individuals from young adults draw support very differently than those in middle or late adulthood. One important factor in having successful social interactions is being able to anticipate and comprehend situations based on the intentions, emotions, beliefs, and desires of others. If we fail to understand a conversation, it can lead to negative social consequences. As we age, our ability to empathize with others can be impacted. Empathy is the capacity to understand and share the feelings of others [17,18]. As defined by Baron-Cohen and Wheelwright [19], empathy is an important ability that allows us to tune in to how someone else is feeling or what they might be thinking. Empathy allows us to understand the intentions of others, to predict their behavior, and to experience an emotion triggered by their emotion. It can be divided into two parts: cognitive and emotional components [20,21,22,23,24]. The cognitive component, also known as perspective-taking, helps us to recognize the emotions of others at a cognitive level. On the other hand, emotional empathy is our ability to feel an emotion similar to another person’s, even though we did not directly experience the event that caused the emotion [24].

In short, empathy plays a central role in social interactions (e.g., pro-social behavior, inhibition of aggressive behavior, and externalizing problem behavior), allowing us to interact effectively in the social world [16,19,25]. Of particular interest is the fact that older adults experience some reduction of empathy [26,27] associated with greater risk for loneliness and depression and poorer personal life satisfaction [28,29]. Chen and colleagues [30] found that older adults reported lower trait empathic concern and personal distress than the younger group. 

It is well-established through research that older individuals generally exhibit lower levels of cognitive empathy when compared to younger individuals. However, it has been observed that older adults tend to display higher levels of emotional empathy [29,31]. Previous studies have shown how cognitive empathy also varies as a consequence of age since older adults (usually older than 64 years) obtain lower scores on cognitive empathy in comparison with younger people (with age ranges between 17 and 56 years) when using performance tests [29,31,32,33]. Research on the development of empathy indicates that emotional and cognitive empathy follow different paths of growth. Cognitive empathy tends to develop at a slower and more consistent pace throughout life, therefore it may require a greater dependence on learning experiences. In contrast, emotional empathy remains relatively stable throughout the developmental process [34,35,36]. According to Labouvie-Vief [37], there is an inverted U-shaped curve in the relationship between cognitive empathy and age. This theory was later confirmed by O’Brien et al. [38] and Gutierrez-Cobo et al. [39]. 

Empathy studies can have inconsistent results due to differences in measurement methods, sample sizes, and gender imbalances. Various tests evaluate different aspects of empathy, and sometimes the measurement technique defines the type of empathy being assessed [40]. There are many measures of empathy, but the most commonly used empathy tests are the Reading the Mind in the Eyes Test (RMET) [41], which primarily measures cognitive empathy capacity, and the Empathy Quotient (EQ) [19], which provides an overall empathy score without dividing between affective and cognitive empathy [29,42].

Empathy is one of the factors underlying social support: individuals with a high capacity for empathy may more actively understand the care and support of others [43]. In this vein, people with high empathy have a stronger ability to obtain support actively and can more actively understand the care and support of others. Emotionally competent individuals have sensitivity to care and support from the outside world. On the contrary, people with low empathy may not care about others’ support and concern for them. This may further affect their attitude toward others. People with high empathy are more likely to perceive support and then exhibit higher prosocial behavioral tendencies [44]. 

Therefore, considering the relationship between age, empathy, and social support, in the present study the idea was to assess whether cognitive or emotional empathy mediates the relationships between age and interpersonal support defined as ‘appraisal’, ‘belonging’, and ‘tangible’, on the one hand, and social support defined in terms of a number of people and level of satisfaction, on the other hand. After controlling for gender and educational level, the hypothesis was formulated as follows: empathy partially mediates the relationships between age and social support and social network. The aim of the study was therefore to better understand the role of empathic capacity in receiving social support taking into account the age of the participants. Based on the hypothesis that the better the empathic capacity the greater the social support received, the idea is to implement psycho-education and awareness-raising programs on the basis of these results to help people improve their mentalization capacity.

## 2. Method

### 2.1. Participants

The eligible study sample was composed of individuals without neurological or psychiatric disorders. To exclude the presence of cognitive decline in participants older than 45 years, the Mini-Mental State Exam (MMSE) (Italian version, [45]) was administered. Additionally, the history or the presence of neurological or psychiatric diseases was investigated by an informal interview carried out before the test phase. Seventeen participants were excluded because they reported having cerebral ischemia or head trauma or having scored below the cutoff of the MMSE (cut-off = 23). A final sample of 441 participants, all native Italian speakers, participated on a volunteer basis; they did not receive any compensation for their participation. All participants came from City Clubs and Workers’ Clubs as well as from the three universities involved in which poster and flyer advertisements were displayed. They had an age ranging from 18 to 91 years (mean age = 42.51, SD = 16.93; age range 18–91 years; 220 males and 221 females) and a full-time education ranging from 5 to 18 years (mean = 13.52 years, SD = 3.06 years). The study was approved by the Department of Psychology, Sapienza, University of Rome, in accordance with the Declaration of Helsinki, and the Committee itself monitored the execution and results. Each participant signed the written consent form after the procedures had been fully explained to them. 

### 2.2. Instruments

#### 2.2.1. Social Support Measures

Interpersonal Support Evaluation List shortened version–12 items (ISEL-12) [46]. It is a 12-item scale investigating three types of social support (appraisal: the perceived availability of someone with whom to discuss issues of personal importance; belonging: the perception that there is a group with which one can identify and socialize; and tangible: the perceived availability of material aid). ISEL-12 showed acceptable reliability and validity. A high level of internal consistency and reliability (Cronbach’s alpha = 0.866) and item homogeneity were confirmed (i.e., refs. [47,48]).

Participants respond on a four-point Likert scale ranging from 1 (definitively false) to 4 (definitively true); on each subscale, the score ranges from 0 to 12. Higher scores correspond to a high perception of social support received. 

Social Support Questionnaire 6 (SSQ6) [49]. It is a 6-item questionnaire developed to measure both the social support network and the satisfaction of the social support received. Each item solicits a two-part answer: Part 1 asks participants to list up to nine people that fit the description stated in the question (e.g., “Whom can you really count on to help you feel better when you are feeling generally down in the dumps?”) and are available to provide support. Part 2 asks participants to indicate for each of the people indicated in the first part, the level of satisfaction using a 6-point Likert scale, ranging from ‘1—very dissatisfied’ to ‘6—very satisfied’. The score reflecting the number of people ranges from 6 to 54, whereas the score related to satisfaction ranges from 6 to 36. The SSQ6 had high internal reliability and correlated highly with the SSQ and similarly to it with personality variables; this is also confirmed in other languages, for instance, the Spanish-language version of the SSQ-6 had an overall Cronbach’s alpha of 0.83 [49,50].

#### 2.2.2. Empathy Measures

Empathy Quotient (EQ) [19]. It is a 60-item questionnaire: 40 items measure empathy, and 20 are filler items. EQ assesses empathy in adults. According to Baron-Cohen and Wheelwright [28], the items are not separated into cognitive and emotional empathy as the two components are often not easily distinguishable as they co-exist. Each item is a first-person statement on a 4-point Likert scale (ranging from Strongly Agree to Disagree Strongly). Baron-Cohen and Wheelwright [19] established a cut-off of 30 to screen for autism spectrum disorders. The instrument is scored on a scale of 0 (the least empathetic possible) to 80 (the most empathetic possible). The EQ has a high level of internal consistency with Cronbach’s alpha of 0.97 [51].

Reading the Mind in the Eyes Test (RMET) ([41], Italian version, [52]). It is composed of 36 photographs of the eye region of 19 actors and 17 actresses surrounded by four single-word descriptors of mental state (e.g., bored, angry). One of these descriptors fit with the mental state depicted in the photo, and the others were incorrect. Participants had to choose the correct mental state for each photo. The RMET has an acceptable internal consistency (see refs. [51,53]). It is based on a four-alternative forced-choice paradigm, with a 25% correct guess rate. The test score was the number of descriptors correctly identified (Maximum score: 36). 

### 2.3. Procedure

Participants were tested individually in a quiet room with artificial lighting and seated on a height-adjustable chair filling in questionnaires after having answered a socio-demographic-anamnestic interview and performing the MMSE.

## 3. Statistical Analysis and Results

Analyses were carried out using IBM SPSS Statistics software v.24 (2016). Data were first transformed into z-scores to transform data from different tests onto the same scale. Then, data were checked for the presence of potential univariate outliers considering a cut-off of ±4 z-scores as the reference values for samples larger than 100 [54,55]. In total, 6 outliers were detected and excluded from subsequent analyses. The new sample consisted of 435 participants. Then, data were tested for normality using the Kolmogorov-Smirnov Test showing that all variables of interest were not normally distributed: Z_Age_ = 0.124, *p* < 0.0001; Z_Educational level_ = 0.257, *p* < 0.0001; Z_ISEL12-Appraisal_ = 0.085, *p* < 0.0001; Z_ISEL12-Belonging_ = 0.124, *p* < 0.0001; Z_ISEL12-Tangible_ = 0.138, *p* < 0.0001; Z_SSQ6-People_ = 0.077, *p* < 0.0001; Z_SSQ6-Satisfaction_ = 0.489, *p* < 0.0001. Therefore, in light of the non-normality distribution of data, Spearman’s Rho correlation analysis was carried out (see Table 1). In general, age correlated negatively to empathy quotient (r = −0.170, *p* < 0.01), ISEL12-Belonging (r = −0.111, *p* < 0.05), and ISEL12-Tangible (r = −0.160, *p* < 0.01), that is, the higher the age, the lower the empathy quotient. Yet, the empathy quotient correlated positively to all scores of ISEL-12 (Appraisal: r = 0.305, *p* < 0.01; Belonging: r = 0.312, *p* < 0.01; Tangible: r = 0.242, *p* < 0.01) and SSQ6 (People: r = 0.136, *p* < 0.01; Satisfaction: r = 0.168, *p* < 0.01) (see Table 1 for details): the higher the empathy quotient the higher the social support. Given that gender and educational level were not correlated to the outcomes (dependent variables), the mediation analyses were carried out without controlling for these covariates. In addition, because RMET did not correlate with any measure of social support, mediation analyses were not conducted using RMET as mediator.

The hypothesis that the empathy quotient mediates the association between age and social support was investigated using the PROCESS macro for SPSS (version 3.5) [56]. Five mediation models were carried out, one for each social support score (see Figure 1), using age as a focal predictor and empathy quotient as the mediator. Bootstrap samples (n = 5000) were used. Bootstrapping is a non-parametric method that bypasses the issue of non-normality distribution [57,58].

As regards the first model, using ISEL-12-Appraisal as the outcome, the direct effect of age was not significant (b = 0.02, *p* = 0.68). Age negatively predicted the empathy quotient (b = −0.18, *p* < 0.001), which in turn positively predicted the outcome (b = 0.21, *p* < 0.001). Therefore, the indirect effect was significant (indirect effect = −0.0373, 95% BootLLCI = −0.0637—BootULCI = −0.0163). 

As regards the second model, using ISEL-12-Belonging as the outcome, the direct effect of age was not significant (b = −0.09, *p* = 0.06). Age negatively predicted the empathy quotient (b = −0.18, *p* < 0.001), which in turn positively predicted the outcome (b = 0.28, *p* < 0.001). Therefore, the indirect effect was significant (indirect effect = −0.0506, 95% BootLLCI = −0.0824—BootULCI = −0.0231). 

As regards the third model, using ISEL-12-Tangible as the outcome, the direct effect of age was not significant (b = −0.05, *p* = 0.28). Age negatively predicted the empathy quotient (b = −0.18, *p* < 0.001), which in turn positively predicted the outcome (b = 0.21, *p* < 0.001). Therefore, the indirect effect was significant (indirect effect = −0.0381, 95% BootLLCI = −0.0651—BootULCI = −0.0162). 

As regards the fourth model, using SSQ6-People as the outcome, the direct effect of age was not significant (b = 0.07, *p* = 0.16). Age negatively predicted the empathy quotient (b = −0.18, *p* < 0.001), which in turn positively predicted the outcome (b = 0.12, *p* < 0.05). Therefore, the indirect effect was significant (indirect effect = −0.0217, 95% BootLLCI = −0.0462—BootULCI = −0.0044). 

As regards the fifth model, using SSQ6-Satisfaction as the outcome, the direct effect of age was not significant (b = −0.07, *p* = 0.14). Age negatively predicted the empathy quotient (b = −0.18, *p* < 0.001), which in turn did not predict the outcome (b = −0.04, *p* = 0.40). Therefore, the indirect effect was not significant (indirect effect = −0.0074, 95% BootLLCI = −0.0067—BootULCI = 0.0249). 

## 4. Discussion

In the present study, we found that the three types of social support (Appraisal, Belonging, and Tangible), as measured by the ISEL-12, and the social support network, as measured by SSQ6, are fully mediated by the empathy quotient and the empathy quotient changes as subjects’ age increases. Several studies demonstrate that empathy in aging is a crucial capacity because it predicts loneliness; specifically, people with poor empathy experience greater levels of loneliness [29,44]. In general, empathy affects the quality of older adults’ relationships [28,44] and life satisfaction [44], which relate to increased morbidity in the elderly. In addition, loss of empathy has been considered a key symptom in patients with Alzheimer’s disease and frontotemporal dementia, and some have suggested that these measures might also help distinguish between the two conditions (e.g., refs. [59,60]). Gouveia et al. [61] found a decline in the EQ’s emotional and social subscales in older adults. 

Another aspect that should be considered when measuring empathy is related to the instrument used. For example, in our study, we used, on one hand, the Reading the Mind in the Eyes Test (RMET) [41], which primarily measures cognitive empathy capacity by inferring complex emotions and other mental states from photographs of the eye region of human faces, and on the other hand, the Empathy Quotient (EQ) [19], which is a self-report questionnaire that provides an overall empathy score without dividing between affective and cognitive empathy [42]. By using RMET and EQ, results are often inconsistent. For example, some studies have reported that older adults have lower cognitive empathy than younger adults [25,26,27,29]; one study found no difference between young and old participants [30]; whereas another study showed that older adults are characterized by higher levels of empathy [38]. Consistent with Schieman and Van Gundy [62], our data showed that both tests correlate negatively with age, indicating lower empathic ability as age increases. 

In the empathy literature, data on the consistency between the two tests are also discordant. In fact, while some studies found a relationship between RMET and EQ [51,63], others did not [52]. It has also been noted that the interpretation of performance during the eyes test is complicated by its dependence on verbal ability [51,64] and the influence of education, race, and ethnicity [65]. This may explain why our data find that only the EQ correlates positively with the measures of social support we used. Furthermore, both the EQ and the measures of social support used in our work, unlike the RMET, are self-report questionnaires that may suffer from social desirability bias, being influenced by demand characteristics, as individuals may want to appear empathetic or self-sufficient because these latter are believed to be desirable characteristics.

In the present study, in agreement with some data from the literature [26,27,29], we found that increasing age decreases cognitive and emotional empathy and that empathy is crucial for people to be satisfied with the social support they receive and perceive. As people age, they often require more assistance from others due to physical and psychological challenges. However, the decline in empathic capacity can be detrimental to their social support network, which may be large and dense. In particular, age does not have a direct effect on perceived social support, but it correlates negatively with empathy, i.e., older adults are less empathic than younger subjects, and in turn, lower empathic abilities mediate the perception of less social support. 

Thus, it can be hypothesized that the changes found in the literature [29,31] in the quality and quantity of social interactions and the composition of social networks were due to an effect mediated by the decreased ability to decode and understand the intentions of others, to predict their behavior, and to experience an emotion triggered by their emotion. In fact, it is conceivable that older adults are still able to fully empathize with people they are closest to and have known for a long time (such as family members) but have more difficulty with new people, with the consequence that they prefer to spend their free time with family members rather than forming new relationships [15]. An alternative hypothesis comes from the study by Richter and coworkers [66], who found that older adults show better performance on tasks that were relevant to them, suggesting that, in general, older adults perform lower than younger adults on tests of empathic accuracy, except when the information is emotionally relevant to them. However, a confirmation of reduced empathy in aging also comes from two neuroimaging studies [30,67]. These studies showed that, despite conflicting behavioral results regarding empathy, older adults show reduced activity in regions typically associated with empathy in younger adults (e.g., anterior cingulate and insula). 

In a review [29] concerning the psychological and neural mechanisms of empathy in aging, it emerged that older adults have lower cognitive empathy (to understand others’ thoughts and feelings) than younger adults but similar and, in some cases, even higher levels of emotional empathy (to feel emotions that are like others’ or feel compassion for them). This aligns with reduced activity in a critical brain area associated with cognitive empathy and supports our results. Indeed, we found that only one test (EQ) correlates with all aspects of social support (appraisal, belonging, and tangible) and the satisfaction of the social support received, probably because EQ measured both cognitive and emotional empathy with respect to RMET. 

From a certain point of view, it may seem a contradiction that as I get older, at the time when I most need to receive social support, I lose the ability to be empathic on a cognitive level, and this has a negative effect on the possibility of receiving support. It is also possible that young people’s self-assessment of support is superficial and not significantly linked to tangible support; however, the fact that the EQ mediates the tangible dimension would seem to refute this interpretation. Understanding others’ perspectives is crucial in social interactions and tends to decrease with age [68]. Henry et al.’s meta-analysis [31] reveals that older adults tend to have weaker Theory of Mind (ToM) skills, regardless of task type (cognitive, affective, mixed) or mode (verbal, visual, static, dynamic). In general, advancing age worsens the ability to understand complex mental states felt by other people.

The results of the present study emphasize that in order to increase the chance of graceful aging, social cognition is a vitally important aspect. Although these data are interesting and undoubtedly useful, the study is not without its limitations. First, the presence of a sample unevenly distributed by age, especially in our sample with few elderly people (over 75 years of age) are the most fragile from a socio-cognitive point of view. Secondly, social support and the social network were measured with a questionnaire that requires the subject to self-assess these two aspects. As all self-assessments, this data can be affected by individual expectations and beliefs as well as personality traits of those who fill out the questionnaire.

## 5. Conclusions and Future Perspectives

In conclusion, the present study demonstrates that the empathy quotient is strictly related to social support. The higher the empathic skills are, the greater the social support received. As our results also point to an essential relationship with advancing age in which participants are less empathetic, this leads us to the importance of providing tools in certain age groups to improve this skill that will also lead the person to a better quality of life by reducing loneliness and increasing the amount and quality of support received.

It is clear from our findings that focusing on psychological and cognitive well-being through cognitive empathy training is crucial. This training helps in enhancing cognitive empathy capacity, which in turn, helps in maintaining social networks and support received. Identifying effective socio-cognitive training approaches for healthy individuals may prevent the development of mental or physical disease and reduced quality of life. 

Empathy is an innate quality, but it is also malleable and can be enhanced by strategic educational interventions [69,70]. Many methods were found to be effective in developing greater levels of empathy, e.g., the use of video recordings, service to disadvantaged communities, reflective writing, and in our view, could easily be used in education to increase empathy in young adults by creating an empathic cognitive reserve that would counteract its decline with age [71].

This is particularly important because it is now well established that some social and environmental variables can have positive and protective effects on the mental and physical health of older adults [72]; in fact, for example, Ricciardi and collaborators [73] found that older adults who had higher levels of perceived social support experienced fewer symptoms of geriatric depression. Continuing to be an active and positive part of the social context brings the older adult several primary and secondary benefits, and through targeted interventions, this is possible. The pandemic taught us that it is possible to be part of a social network even at a distance and showed us how those who managed to maintain social relationships even during the period of home restriction suffered less from the negative consequences of isolation and reduced the psychological phenomena of unease and fear that the condition experienced at that time had brought. Therefore, every intervention aimed at reducing mental health vulnerability is mandatory.

## Figures and Tables

**Figure 1 healthcare-11-02464-f001:**
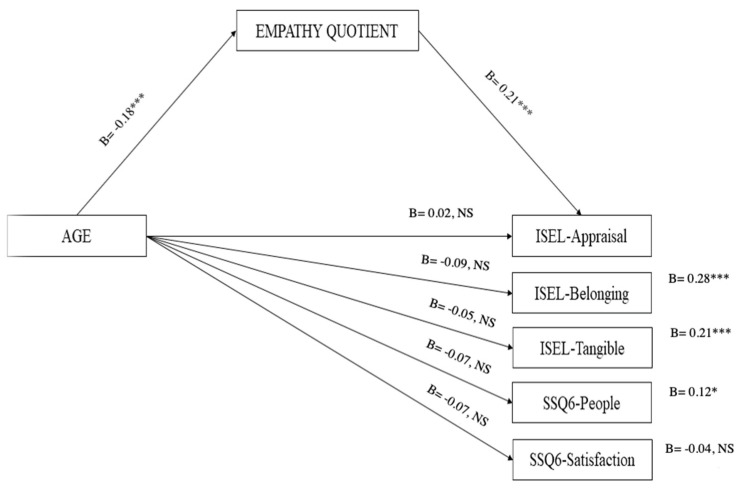
Template of the mediation analysis. Note: *** *p* < 0.001; * *p* < 0.05.

**Table 1 healthcare-11-02464-t001:** Spearman’s Rho correlation between Demographic data, Social Support Measures and Empathy Measures.

	Mean (SD)	Age	Gender	Edu.	EQ	RMET	ISEL-12-A	ISEL-12-B	ISEL-12-T	SSQ6-P	SSQ6-S
**Age**	42.17 (16.75)	1									
**Gender**	---	−0.017	1								
**Edu**	7.45 (3.02)	−0.194 **	0.014	1							
**EQ**	44.24 (12.04)	−0.170 **	0.161 **	0.066	1						
**RMET**	21.94 (5.10)	−0.186 **	0.088	−0.038	0.068	1					
**ISEL-12-A**	12.61 (2.81)	−0.093	0.093	0.007	0.305 **	0.024	1				
**ISEL-12-B**	12.63 (2.32)	−0.160 **	−0.035	−0.021	0.312 **	0.023	0.502 **	1			
**ISEL-12-T**	13.13 (2.23)	−0.111 *	−0.027	−0.020	0.242 **	−0.015	0.565 **	0.586 **	1		
**SSQ6-P**	22.42 (10.81)	0.011	0.028	−0.017	0.136 **	0.017	0.253 **	0.090	0.143 **	1	
**SSQ6-S**	7.45 (7.49)	0.089	0.076	−0.008	0.168 **	0.091	0.265 **	0.241 **	0.225 **	0.193 **	1

Note: ** *p* < 0.01 (two-tailed); * *p* < 0.05 (two-tailed); Edu = Educational Level; EQ = Empathy Quotient; RMET = Reading the Mind in the Eyes Test; ISEL-12-A = ISEL-12-Appraisal; ISEL-12-B = ISEL-12-Belonging; IESL-12-T = ISEL-12-Tangible; SSQ6-P = SSQ6-People; SSQ6-S = SSQ6-Satisfaction.

## Data Availability

Data may be asked to the corresponding author.

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
