# Peer review of "The Key Role of Empathy in the Relationship between Age and Social Support"

_healthcare, 2023, doi:10.3390/healthcare11172464_

Round 1

Reviewer 1 Report

Authors should improve

1. adjust the title to the contents presented.

2. Define what they understand by empathy 

3. Define the objective of the study

4. Explain the criteria for the selection of participants and the application of the tests. 

5.  Separate the conclusions in the discussion 

6. Create a conclusion section 

7. Establish future proposals

Reviewer 2 Report

Thank you for this interesting study. There are a few items that, if addressed, would improve the readability of your manuscript. 

1. Please have someone whose first language is English edit the manuscript.

2. Your introduction is very long and detracts from your results. Consider shortening the introduction.

3. Your reference list is impressive but it detracts from the results. Consider decreasing the number of references and get to your research more quickly, with fewer distractions.

4. Can you describe your instruments more succinctly or put the more complete descriptions in an appendix?

5. Why is empathy not one of your key words?

6. It is considered to be more polite to use the phrase older adults instead of elderly people or older people. 

7. The MMSE may be a well established instrument to measure cognition but it is not as sensitive as other instruments. Consider using other assessment tools int he future. 

8. In line 283 you refer to "large e dense social support system".  What is  large e dense?

9. Line 145: change Liker to Likert.

10. You describe a large sample size with a 73-year span in age.  Did you have equal numbers of individuals per decade of age? Was the impact of age linear?  Or, for example, did people with young children show more empathy for a while, perhaps until they became empty nesters? 

11.  If you take more space (which you can do if you reduce the length of the introduction and the number of references) to describe your data analyses, the reader will understand better how you reach your conclusions.

This manuscript is difficult to follow because it looks like it was written in Italian and then translated. It needs a good edit.

Reviewer 3 Report

Thank you for submitting your research study on empathy and social support in older adults for publication consideration. It is a fascinating topic and you clearly describe the importance of the study. I was wondering if you could expand on the reliability and validity of the instruments used to measure the major constructs in your study. Do you have reliability and validity data to support the use of these instruments? Also, what recommendations do you have for additional research in this area?  What are the major limitations of your study? Also, what was the ethnic/racial composition of your study population?  Finally, I noticed that only about 14 of the 95 references are within the last five years. Has much research been done in this area recently? 

The manuscript could benefit from editing of English language. It reads a little rough.

Round 2

Reviewer 1 Report

The article can be published 

Reviewer 3 Report

Thank you for making the suggested revisions. I believe it significantly improves the quality of the manuscript and am pleased to now recommend it for publication.

Minor edits could still benefit the manuscript but it has improved since the first version.